# Nocturne: a scalable driving benchmark for bringing multi-agent learning one step closer to the real world

**Eugene Vinitsky**[*†]
Meta AI, UC Berkeley
vinitsky.eugene@gmail.com

**Nathan Lichtlé**[*]
UC Berkeley
École des Ponts ParisTech
nathan.lichtle@gmail.com

**Xiaomeng Yang**[*]
Meta AI
yangxm@fb.com

**Brandon Amos**
Meta AI
bda@fb.com

**Jakob Foerster**
Univeristy of Oxford
jakob.foerster@eng.ox.ac.uk

## Abstract

We introduce *Nocturne*, a new 2D driving simulator for investigating multi-agent coordination under partial observability. The focus of Nocturne is to enable research into inference and theory of mind in real-world multi-agent settings without the computational overhead of computer vision and feature extraction from images. Agents in this simulator only observe an obstructed view of the scene, mimicking human visual sensing constraints. Unlike existing benchmarks that are bottlenecked by rendering human-like observations directly using a camera input, Nocturne uses efficient intersection methods to compute a vectorized set of visible features in a C++ back-end, allowing the simulator to run at 2000+ steps-per-second. Using open-source trajectory and map data, we construct a simulator to load and replay arbitrary trajectories and scenes from real-world driving data. Using this environment, we benchmark reinforcement-learning and imitation-learning agents and demonstrate that the agents are quite far from human-level coordination ability and deviate significantly from the expert trajectories. Code for Nocturne is available at https://github.com/facebookresearch/nocturne.

## 1 Introduction

This paper presents *Nocturne*, a new simulator and benchmark for multi-agent driving under human-like sensor uncertainty that is intended to aid the process of studying real-world multi-agent coordination and learning. Instead of combining the challenges of coordination with feature extraction from images, Nocturne is a 2D simulator that generates vector representations of the set of objects and road points that would be visible to an idealized human driver (see Fig. 1 for an example) and supports head-tilt to acquire additional information about blind spots. In contrast to driving benchmarks that achieve partial-observability by using a camera input, Nocturne uses efficient visibility-checking methods and a C++ back-end to enable us to construct observations and step the dynamics of a single agent at thousands of steps-per-second (see Appendix Sec. F for an exact analysis). This speed is key to its use in multi-agent learning settings where frequently billions of environment interactions are needed to learn performant agents [3, 20]. In contrast to many existing multi-agent learning benchmarks, Nocturne is neither zero-sum nor fully-cooperative but mixed-motive, combining the challenges of coordination and cooperation.

---

[*]**These authors contributed equally to this work.**
[†]Corresponding Author

36th Conference on Neural Information Processing Systems (NeurIPS 2022) Track on Datasets and Benchmarks.

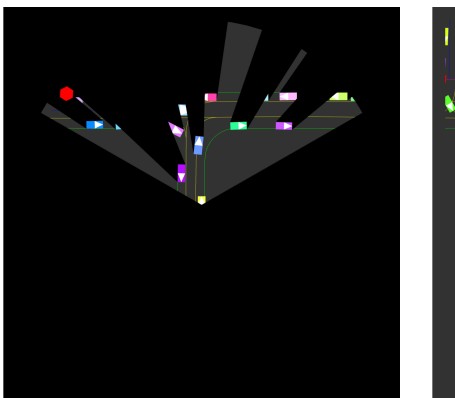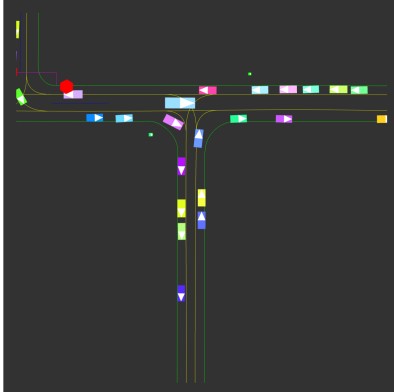

Figure 1: A visual depiction of the obstruction model used to represent the objects that are visible to the agents. (Left) Obstructed view with the viewing yellow agent in the center of the cone. (Right) Original scene centered on the cone agent with an unobstructed view.

Crucially, Nocturne is not just a simulator. Instead, it is built upon open-source driving data and features a diverse set of real-world scenes (see Fig. 2) that probe the ability of agents to safely navigate and coordinate in complex scenes such as intersections, roundabouts, parking lots, and highways. We use this data as a source of experts for imitation, to flexibly vary the number of controlled agents in the scene, and as a train-test split for validating the generalization ability of a human-driver model.

The challenge we propose is to learn (or otherwise design) policies that achieve the same set of final states as human experts (throughout this work we will call this the *goal rate* and the final state the *goal position*) while achieving a 0% collision rate. Achieving a high goal rate alongside a negligible collision rate is challenging for any policy design scheme (both learning and non-learning) due to the combination of the high-dimensional state space, the partial observability of the scene, the large number of interacting agents, and the decentralization of the policies at test time. The secondary challenge in Nocturne is to find policies that closely mimic human behavior at the trajectory level. Building on top of the human data we provide an evaluation scheme and off-the-shelf baseline implementations including evaluations.

We provide a guide to the construction and rules of the benchmark as well as results on using Nocturne to design agents and test their capabilities and human-similarity. We demonstrate that learning effective agents in Nocturne is challenging; tuned RL and imitation learning baselines struggle to successfully complete the highly interactive scenes. Finally, we demonstrate that the agents achieve relatively low distance to the expert trajectories and show in Appendix Sec. B that there does not appear to be a zero-shot coordination problem [16] at this level of agent capability.

## 2 Related Work

**Multi-agent traffic simulation tools and benchmarks:**
In terms of multi-agent driving benchmarks that require both acceleration and steering control, there are several closely related benchmarks that differ in terms of ease of implementing multi-agent interaction, being data-driven, 2D vs. 3D, the mechanism by which they support partial observability, or the rate at which they can construct the partially observed state for an agent. We summarize some of these differentiating features in Table 1.

The closest works to ours are BARK [5], SMARTS [42], and MetaDrive [22]. SMARTS [42] supports multi-agent driving in a wide variety of interactive driving scenarios and offers a large set of default human driving behaviors to use for testing autonomous vehicles. BARK [5], like Nocturne, is a 2D goal-driven simulator with support for external datasets and contains a wide variety of large-scale scenarios and multi-agent support. Finally, MetaDrive [22] also supports real-world data and multi-agent interaction and is able to achieve 300 FPS image rendering by rendering lower fidelity images. The main differentiating feature from these works is the support for acquiring the set of objects that are visible without requiring the rendering of camera images; to our knowledge Nocturne is the only available simulator that can jointly compute an agent's visible objects and the agent's dynamics

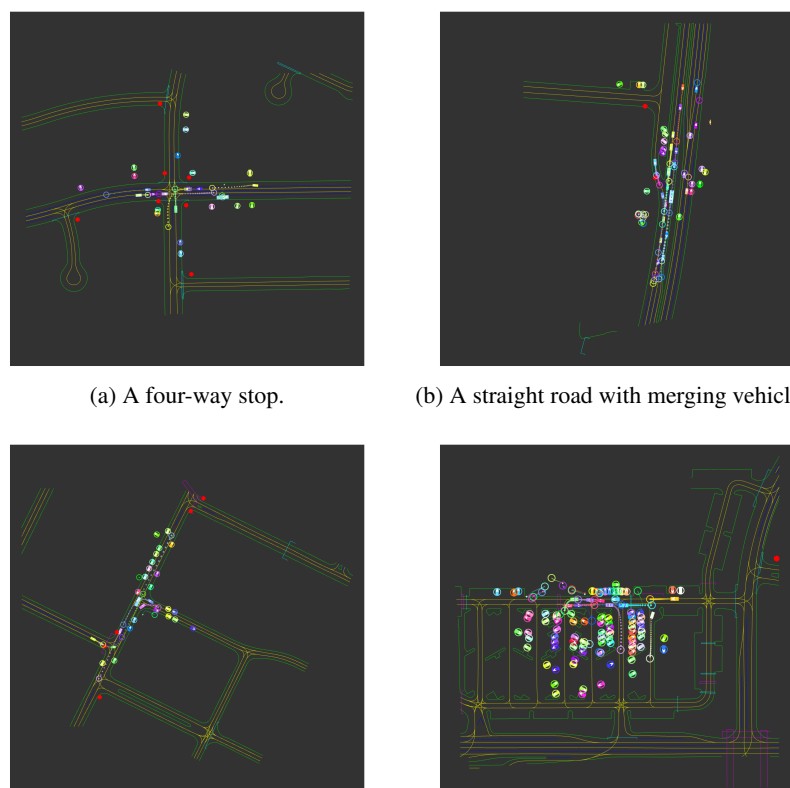

(a) A four-way stop.    (b) A straight road with merging vehicles.

(c) Unprotected turns with conflicting routes.    (d) A crowded parking lot.

Figure 2: Four scenes demonstrating the diversity of the navigable scenes in Nocturne. Colored circles represent the goal position of the corresponding colored agent. Dots represent the trajectory of the agent, with opacity increasing as time goes on. Videos of experts negotiating these scenes can be found at nathanlct.com/research/nocturne.

step above 2000 steps-per-second (see Appendix Sec. F for exact details). Additional simulators are summarized in Table 1.

**Partially observed multi-agent benchmarks:**
There are a wide variety of partially-observable multi-agent benchmarks not focused on driving. Within the card-playing domain, Hanabi [3] is frequently used to investigate coordination under partial observability and features between 2-5 agents. The Starcraft multi-agent benchmark (SMAC) [32], perhaps the most ubiquitous MARL benchmark, features many agents and high-dimensional observations but does not come with human data and many algorithms now achieve perfect performance in this challenge [40]. Melting Pot [20] features a huge diversity of many-agent mixed-motive challenges but does not have available expert data. Finally, games such as Poker, Stratego, and Diplomacy are frequently studied.

**Open-Source Trajectory Data:**
We provide a brief overview of large driving datasets that contain both map information and annotated trajectory data for vehicles, pedestrians, and cyclists. Available trajectory data can be categorized along the size of the dataset, the diversity of data in the dataset, the method of collection, and the available annotations of the scene (e.g. maps, traffic lights, road object trajectories). This work uses the Waymo Motion dataset [12] as it has a high diversity of scenes across many cities and contains annotations for cyclists, pedestrians, and traffic lights, features that will be used in future versions of Nocturne. However, we note that Argoverse 2 [38] and nuScenes [7] have similar features and could have been used; nuScenes also has complete traffic light annotations instead of annotations only for ego-observed traffic lights. The Lyft Level 5 dataset [15] appears to be the largest available dataset and has thorough road and vehicle annotations but is drawn from a single stretch of road. For non-egocentric datasets, the INTERACTION dataset [41] contains a diversity of scenes across

Table 1: Comparison of representative driving simulators. State-based partial observability refers to whether the set of visible objects can be queried. Expert data refers to whether recorded trajectory data is available for the scenes. Baseline human models refers to whether explicit models of human driving are available to control road objects in the scene.

| Simulator | Multi-agent Support | 2D/3D | State-Based Partial Observability | Expert Data | Baseline Human Models |
|---|---|---|---|---|---|
| CARLA [11] | | 3D | | | ✓ |
| SUMMIT [8] | ✓ | 3D | | | ✓ |
| MACAD [26] | ✓ | 3D | | | ✓ |
| Highway-env [21] | | 2D | | | ✓ |
| Sim4CV [23] | | 3D | | | |
| Duckietown [28] | | 3D | | | |
| SMARTS [42] | ✓ | 2D | | | ✓ |
| MADRaS [33] | ✓ | 2D | | | ✓ |
| DriverGym [19] | | 2D | | ✓ | ✓ |
| DeepDrive-Zero [30] | ✓ | 2D | | | |
| MetaDrive [22] | ✓ | 3D | | ✓ | ✓ |
| VISTA [1] | ✓ | 3D | | ✓ | ✓ |
| **Nocturne** | ✓ | 2D | ✓ | ✓ | |

both cities and countries, thereby capturing a variety of driving styles and norms, but is an order of magnitude smaller than other datasets.

## 3    Benchmark construction

### 3.1    Defining a Nocturne Scene

In the following sections, we will refer to objects that can move (vehicles, pedestrians, cyclists) as *road objects* and anything that cannot move (lane lines, road edges, stop signs, etc.) as *road points*. *Road points* are connected together to form a polyline. We will refer to the type of road polyline that should not be crossed by vehicles as a *road edge*.

Nocturne scenes require a map consisting of polylines, a set of initial and final road object positions, and optionally a set of trajectories for the road objects. Nocturne currently acquires its scenes, goals, and expert trajectories from the Waymo Motion dataset [12] but can be easily configured to support any dataset that represents its road features as points or polylines. The Waymo Motion dataset consists of $487004$ nine-second trajectory snippets discretized at a rate of 10 Hz with the first second intended to be used as context and the latter eight seconds to be used for prediction.

One challenge of selecting and constructing the scenarios for Nocturne is that these trajectories are collected by labeling the vehicles observed by a Waymo car as it drives. Consequently, we do not have a complete birds-eye view of the scene. Hence, there are cars that may have been in the scene that were not visible to the Waymo vehicle and therefore are not included in the dataset; for the same reason, the expert trajectories are also incomplete and may not persist throughout the entire duration of the rollout. In other words, the expert trajectories contain agents that unpredictably flicker in and out of existence. Similarly, the set of traffic lights that were visible to the Waymo vehicle may be insufficient to uniquely determine the underlying traffic light state. For this reason, this first version of the Nocturne benchmark comes with a few restrictions: we do not use traffic lights and we only include vehicles available at the first time-step of the scene to avoid collisions induced by unpredictable vehicle appearances. We filter out scenes that contain traffic lights which leaves the majority of the remaining scenes as roundabouts, unsignalized intersections, arterial roads, and parking lots. This leaves the benchmark consisting of $134453$ snippets.

## 3.2 Partial Observability Model and Collision Handling

To support the investigation of coordination under partial observability, agents in Nocturne come with a configurable view-cone as is shown in Fig. 1. An element (road object or road point) is considered observable if there is a single ray in the cone that intersects with that element that does not pass through any road object on its path to the element. Stop signs are always visible if they are within the agent view-cone even if they would be obscured on the assumption that they would be raised at a sufficiently high level. We select the angle of the cone to be 120 degrees (approximately the range of binocular vision) and 80 meters radius. We note that this does not perfectly mimic a human visual model which has additional complexities such as dynamic variations in the functional field of view [10], phenomena relating to interactions between visible objects such as crowding [6], and the necessary ability to pay attention to objects further than 80 meters away at high speeds. Furthermore, we currently represent our vehicles as rectangles and so this model of partial observability will be overly pessimal and neglect situations such as the ability for a driver to sometimes see over the hood of a car or 3-D effects like tall cars seeing over small ones.

The primary challenge in computing which road points and objects are observable is their relatively large number: a scene contains about 30 vehicles and 4800 road edge points on average. We use a Bounding Volume Hierarchy (BVH) to maintain the road objects and select candidates for potentially observed vehicles. We build the BVH using approximate agglomerative clustering [14] to generate a high-quality BVH. When computing the observed objects, we first use the BVH to select the candidates that lie in the *axis-aligned bounding box* (AABB) of the conic view field. Since there are a comparatively larger number of all types of road points (about 16000 on average), we use a 2D range tree [4] to maintain all of the road points. When computing the observed road points, we do a range search in the 2D range tree to select the candidates that lie in the AABB of the conic view field. For both vehicles and road points, once we have the candidates in the conic view-field, we perform a brute-force visibility check for the object and road points respectively by ray-casting from the viewing agent to all the candidate vertices. Finally, these data-structures are similarly used to accelerate collision checking.

## 3.3 Construction of the Partially Observable Stochastic Game

**Definition of the state space**

Nocturne supports two possible state representations: a rasterized image and a vectorized representation of that image. While we provide default state representations, we consider it fair game on the benchmark to use any other representation as long as only objects that are visible under the conditions in Sec. 3.2 are presented to the agent. Conforming to these consistent rules about visibility allows for a fair comparison between different algorithmic approaches.

For the default vectorized representation, we adopt a fully descriptive set of features (speed, angle, width, length, etc.) for the road objects and use the VectorNet [13] representation for the road points. We will refer to the vehicle whose observation is being returned as the *ego vehicle*. Note that by default all features that can be placed into relative coordinates are returned in relative coordinates to the ego vehicle (e.g. speed is relative speed, heading is relative heading, etc). The agent goal is set to be the final position, speed, and heading of the expert agent. An agent is considered to have achieved its goal if it is within 1 meter of the final position and within $1 \frac{m}{s}$ of the final speed of the agent and $0.3$ radians of the final heading when the target position tolerance is achieved. For exact observations for the ego object, road objects, and road points, as well as the padding mechanism used to handle variations in the number of observed objects / points / stop signs see Appendix Sec. A.

**Action Space**

Vehicles are driven by acceleration and steering commands that are passed to a bicycle model to update the vehicle state. In the experiments used in this paper we use 6 discrete actions for acceleration, 21 discrete actions for steering, and 5 discrete actions for head tilt with the acceleration actions uniformly splitting $[-3, 2] \frac{m}{s^2}$, the steering actions between $[-0.7, 0.7]$ radians, and the head tilt between $[-1.6, 1.6]$ radians. For more details on the vehicle model, see Appendix Sec. D.

**Environment Dynamics and Goals**

The total length of the expert data is 9 seconds, discretized into steps of size 0.1 seconds. For the first 1 second of the episode, all vehicles obey the expert policy. This is used to construct a history of observations for each agent that can be used to initialize or warm up the policy. After this transitory period, the episode continues for a fixed length of $T = 80$ steps. Agents are provided with a target position and target speed that are taken from the final speed and position of the expert trajectories. If an agent achieves their goal they receive a reward of $T$ and are removed from the system. A vehicle will also be removed if it collides with any road edge or object.

In addition, there is a process for selecting the set of vehicles that are controlled in the environment. First, we only control vehicles that at some point have a speed above $0.05\frac{m}{s}$ and that are more than 0.2m from their goal; in general, these are vehicles that need to move to get to their goal. From this set of vehicles, we remove all vehicles that are already at their goal. Next, we randomly select up to a maximum of 20 of them and set the remainder to replay expert trajectories. This latter process for keeping a maximum number of controlled vehicles is used in this paper due to constraints of the RL library and is not an aspect of the benchmark.

### 3.4 Rules of the Benchmark

We outline here a few rules that we expect solvers to respect to ensure consistency between solutions.

- The size of the view cone is fixed to 120 degrees and a distance of 80 meters. This ensures consistency between the level of partial-observability each controller must handle. Users can also tilt the head of the driver by up to 90 degrees in either direction to acquire more information.

- Only the first 1 second of the trajectory can be used as context or to warm-start a memory-based controller; control of the vehicles must start at 1 second into the trajectory.

- A trajectory that successfully reaches the goal is only considered valid if it reaches the goal within the 8 second time-window. All the scenes are easily solved without a time-constraint by simply creeping forwards slowly.

- The environment comes with default rewards and observations but any amount of reward-shaping or observation sharing at training time is valid. However, at test time only information that is directly observable to the agent can be used as input to the policy. For example, sharing information about other agents' goals would be valid at train time but not at test time.

- Adding additional map information to the agent state space beyond the information provided by default is valid. This corresponds to humans often having knowledge about map layout beyond what is within eyesight.

- The bounds on the action space should be respected: the acceleration should be bounded between $[-6, 6] \frac{m}{s^2}$, the change in heading should not be faster than 40 degrees per second, and driver head tilt should be maintained within 90 degrees in either direction. This rules out solutions that rapidly get to their goal by using accelerations that are outside of possible vehicle speed bounds or excessively sharp turns.

We note that these rules may make some of the scenes unsolvable. For example, real human drivers are not randomly initialized into a scene with a maximum of 1 second of prior context; this context may be critical to safely navigating the scene under the time constraints. Additionally, our model of human perception is not an exact match for true human perception which can extend well beyond 80 meters under certain circumstances. It is possible that there may be missing context at this viewing distance which is crucial for safe navigation. We view these constraints as similar to the challenge of label noise in supervised datasets; our constraints may place an upper limit on the percentage of goals that can be achieved but the benchmark still constitutes a valid comparison between methods as long as the constraints are respected.

### 3.5 Unusual features of the Benchmark

For completeness, we note a few oddities of the benchmark that we believe are critical for potential solution designers to be aware of. These challenges are often properties of labeling noise and the egocentric view under which the data was collected.

**Filtered out pedestrians, cyclists, and traffic lights**

While the dataset contains scenes with traffic lights, pedestrians, and cyclists, the first set of Nocturne environments, NocturneV1, operates solely on scenes that have been filtered to not include traffic lights. Additionally, when constructing the environment, we remove all pedestrians and cyclists from the scene. There is still an option to include them in the visible state, which we do for behavioral cloning. Future versions of the benchmark will consider joint learning of pedestrians, cyclists, and vehicles but for NocturneV1 we only consider vehicles as appropriate modeling of pedestrian and cyclist dynamics is challenging.

**Egocentric data collection**

The Waymo Motion dataset that forms the basis of the first version of Nocturne is collected by driving a sensor-equipped car and recording the trajectories of all visible vehicles. Thus, in the original data vehicles may have trajectories that are shorter than the full 9 seconds and may only appear midway through the trajectory or appear and disappear throughout the trajectory when they are obscured from the view of the sensing car. Rather than suddenly teleport cars into the scene midway through an episode, we choose to only use cars that have valid states beginning of the episode. This reduces the total number of vehicles that might appear in the episode but does not change the feasibility of any of the agent goals.

**Infeasible goals**

Roughly 3% of the vehicles in the dataset have an expert trajectory that crosses an impassible road edge. This is due to labeling errors in the dataset where, for example, small gaps in road edges are occasionally missed that make the road edge crossable. To ensure a benchmark where all goals are achievable, we compute all trajectories where crossing a road edge was necessary to achieve the goal and set these vehicles to replay their expert trajectory rather than be controlled; this corresponds to 3% of all vehicles. Finally, 2% of vehicles in the dataset are initialized in a colliding state. This primarily occurs in parking lot scenes where a vehicle slightly overlaps with the boundary of a parking spot or a nearby vehicle. These vehicles are also removed. For the calculations of these statistics, see Appendix Sec. E.

**Illogical goals**

Occasionally, agents may have goals that seem illogical; for example, agents may be asked to come to a full stop in the middle of a highway. While these goals may seem odd, they are actual states that were achieved by humans driving on the roadways. These odd goals are often a consequence of unobserved objects such as drivers queued up that make the vehicles stop so as to avoid a collision that was not observed by the egocentric data collection process.

## 4  Experiments Setup

We run a reinforcement learning and an imitation learning baseline to demonstrate that these tasks do not appear to be easily solved even after billions of steps. We also briefly investigate whether the policies appear to have a zero-shot coordination challenge wherein policies perform well when paired with agents from their seed but are incompatible with policies from a different training run.

We test the following methods:

- APPO [29] trained in multi-agent mode with a shared policy i.e. every agent is controlled by the same policy but in a decentralized fashion.
- Behavior Cloning [2, 34].

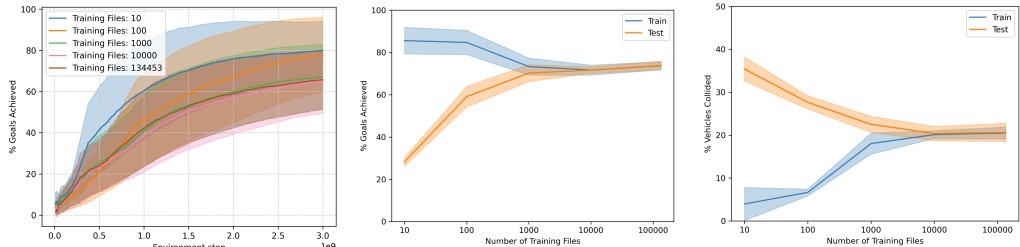

Figure 3: (Left) Success at getting to the specified goal on the training data as a function of number of environment steps. "Training Files: X" means the agent was trained on X fixed scenes sampled from the training dataset. (Middle) Percent of agents that achieved their goals. (Right) Percent of agents that collided.

For the RL method, in addition to the fixed-bonus for achieving the goal, we add the following dense reward to encourage the agent to make progress toward the target position, speed, and heading:

$$r_t = 0.2 \times \left(1 - \frac{||x_t - x_g||_2}{||x_0 - x_g||_2}\right) + 0.2 \times \left(1 - \frac{||v_t - v_g||_2}{40}\right) + 0.2 \times \left(1 - \frac{f(h_t, h_g)}{2\pi}\right) \quad (1)$$

where $x_t$ is the position at time-step $t$, $x_g$ is the goal position, $v_t$ is the speed at time-step $t$ and $v_g$ is the target speed, $h_t$ and $h_g$ are the current and target heading in world coordinates (i.e. not in a relative frame). $f(h_t, h_g)$ returns the minimum angle between $h_t$ and $h_g$. Since coordinates are in a relative frame and the agent cannot observe the world frame, note that $f(h_t, h_g)$ is an observation provided to the agent. The use of this dense reward, which certainly affects the form of the optimal policy, is not a necessary component of the benchmark and here is just used to generate good policies.

All experiments are evaluated on the same 200 randomly selected scenes from the validation set and statistics are computed over an 8 second rollout. The width in all the plots represents twice the standard error. For more details and hyperparameters for both methods and architectures, see Appendix Sec. A.

## 5 Results and Analysis

### 5.1 Success rate of baselines

Fig. 3 shows the performance of the agents on the training set after $3e9$ steps which takes approximately two days on 1 GPU and 10 CPUs. Each line corresponds to a policy trained on a fixed subset of the training scenes. We score our policies in two primary ways: the fraction of vehicles that achieve their goal (goal rate) and the fraction of vehicles that collide with another vehicle or road edge (collision rate). Note that this use of collision rate differs from its use in papers such as [18, 35] which compute collision rate as the fraction of scenes that contain a vehicle-vehicle collision.

We investigate the effect of the size of the dataset on the train and test performance. In the right half of Fig. 3 we can see that the inclusion of a larger training dataset decreases the performance of the algorithms up to 1000 files as they struggle with the diversity of the data. However, the inclusion of additional data narrows the magnitude of the train and test gap and closes it fully at 10000 files. However, it is still possible that a divergence between train and test might re-emerge as agents become more capable on the train set and approach a 100% goal rate.

Finally, Table 2 compares the APPO and BC agents. For the APPO agent, we include the results for the agent trained on 10000 training files. The expert playback row refers to replay of the expert trajectories.

### 5.2 Human-agent trajectory similarity

We analyze the results of our experiments with respect to how human-like the resultant policies are using displacement error between expert and agent trajectories as our metric (i.e. L2 distance between the agent and expert trajectories). To align with the definition used in other works [18, 35] we disable

Table 2: Overview of metrics across methods for an 8 second rollout.

| Algorithm | Collision Rate (%) | Goal Rate (%) | ADE (m) | FDE (m) |
|---|---|---|---|---|
| Expert Playback | 4.9 | 100 | 0 | 0 |
| APPO | $20.3 \pm 0.8$ | $71.7 \pm 0.7$ | $3.1 \pm 0.2$ | $6.1 \pm 0.3$ |
| BC | $38.2 \pm .1$ | $25.3 \pm 0.1$ | $5.6 \pm 0.1$ | $9.2 \pm 0.1$ |

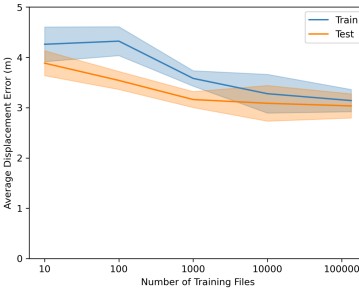 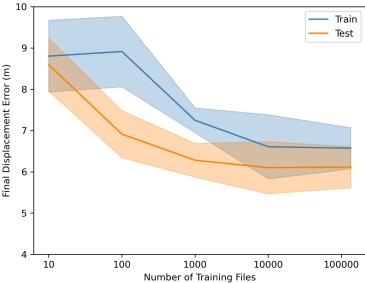

Figure 4: (Left) Average displacement error (mean l2-distance between an agent and an expert at each time-step). (Right) Final displacement error (l2-distance between an agent and an expert at the final time-step that an expert has a valid state).

the removal of vehicles upon collision / reaching the goal. However, we note a few dissimilarities that make comparisons with other works difficult. First, agents are provided with a goal position, a feature that is often not available to other predictive methods. Second, our experts are stepped in scenes that may contain pedestrians or cyclists but our agents are replayed in the same scene without the corresponding pedestrians or cyclists. This can make the magnitude of the displacement error not directly comparable to the values in other works; the low value of the displacement error may simply indicate that a majority of the scenes have unique optima. Fig. 4 examines the average difference in position between the agent and expert trajectories averaged across 5 training runs and demonstrates that the influence of more training data on displacement error is flat after 1000 files.

### 5.3 Policy Failure Modes

Here we investigate the mechanisms under which our policies fail to achieve their goals and collide to shine a light on potential avenues for improvement. The key failure modes we qualitatively observe are failures in scenes in which agents are required to interact with another agent either by waiting or merging. Videos of some of the failure modes can be seen at nathanlct.com/research/nocturne. While we cannot measure interactivity directly, we measure a proxy by looking at the intersection of the expert trajectories. We play the experts forwards, record their trajectory as polylines, and consider a vehicle to have as many interactions as there are intersections of the vehicle's expert trajectory polyline with other vehicle's expert trajectory polylines (i.e. if two vehicles' trajectories in time cross at a point, that's an interaction). This captures interactions such as crossing at a four-way stop, merges, and others but neglects interactions such as complementary left turns and also may unintentionally pick up behaviors such as driving behind another vehicle. Close to 25% of vehicles have at least one interaction. The collision and goal rates as a function of interactions are plotted in Fig. 5 and demonstrate that the goal rate declines precipitously and the collision rate increases sharply as the number of interactions increases. This suggests that our agents have learned to get to their goals but perform poorly in settings where getting to the goal requires coordination with another agent.

## 6 Conclusion

We have introduced Nocturne, a simulator and benchmark intended to aid in the study of human-like decentralized coordination for driving systems. We present results on the applications of RL and imitation to this system, however, there still remains work to be done to build agents that operate with the collision and goal rate that humans achieve as well as how to learn these agents efficiently.

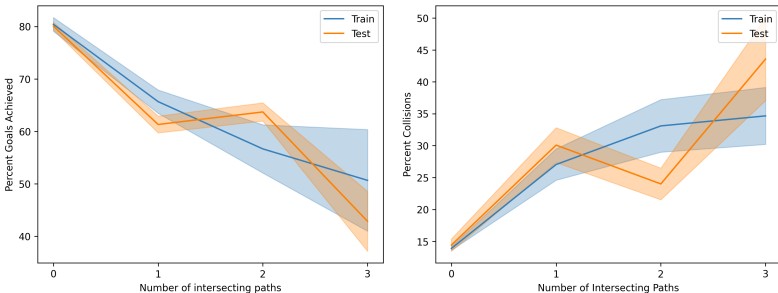

Figure 5: Goal rate (left) and collision rate (right) of vehicles as a function of the number of times that their corresponding expert trajectory intersected with another expert trajectory (intersections). As more than 3 interactions are rare, creating noisy statistics, scenes with more than 3 interactions are placed into the 3 interaction bin.

Given better agents, human-like rules and conventions may be emergent properties of driving safely in these settings [25].

There is also ample remaining work to be done on new benchmarks. Due to the ego-centric data collection, we are forced to remove vehicles once they achieve their goal (i.e. the last observed position of the driver in the data). However, it may be possible to use generative models or other generative mechanisms to sample new goals for the agents to continue their trajectory once they achieve the goals set out in the data. Similarly, generative models could be used to complete the traffic light states and enable the inclusion of the traffic light scenes.

Finally, one open question is how to use Nocturne agents as predictive models of human driving. At the moment a Nocturne agent requires a goal to which it is driving. To use Nocturne agents for prediction (say for an autonomous vehicle trying to predict the motion of agents in the scene), a method for inferring goals from the 1-second context needs to be implemented. A topic for future work is to use supervised methods to predict the goals and enable fully decentralized prediction of Nocturne agents from egocentric observations.

## 7    Reproducibility and Ethical Statement

We do not release trained models as the Waymo Motion dataset restricts the release of trained models. While the dataset contains images that have been anonymized, only trajectory data is used in this work. The benchmark and all files needed to run it are publicly available.

## 8    Acknowledgements

We thank the Python community [37] for creating the core tools that enabled our work including Hydra [39], Pytorch [27], Matplotlib [17], and numpy [24, 36]. A huge thanks to Aleksei Petrenko for a ton of support in tuning and debugging SampleFactory [29]. Thanks to Scott Ettinger for help in understanding some of the peculiarities of the Waymo Motion dataset [12]. Thanks to Rachit Singh for help with some of the results analysis scripts. We would also like to thank the International Emerging Actions project SHYSTRA (CNRS) for support of Nathan Lichtlé.

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
