# A  Experiment Details

## A.1  Architecture and computational resources

The APPO experiments are each run for two days on 1 V100 GPU and 10 CPUs (Intel Xeon CPU E5-2698 v4 @ 2.20GHz) and each experiment is run for 5 seeds. For the RL experiments we run 5 conditions over 5 seeds for 2 days leading to a total of 1200 GPU hours and 12000 CPU hours. For the Imitation Learning experiments we run 5 seeds for three hours leading to 15 GPU hours and 100 CPU hours.

The architecture for the APPO agent is a two-layer neural network with 256 hidden units and Tanh non-linearities followed by a Gated Recurrent Unit [9] with 256 hidden states. The Behavior Cloning experiment is run on a single GPU for five seeds, trained on 1000 files, for 600 gradient steps with a batch size of 512. Here instead of using a recurrent neural network we stack the current state and the prior 4 states as an input and pass this through a three-layer (1025, 256, 128) neural network that then outputs two categorical heads, one for acceleration and one for steering. The acceleration head is binned into 15 bins equally spaced between $[-6, 6]\ \frac{m}{s^2}$ and steering to 43 equally spaced bins between $[-0.7, -0.7]$ radians. As there is no corresponding head tilt in the expert actions we extend the angle of the visibility cone to $\pi$ radians. Note that this violates a rule on the cone shape set forth in Sec. 3.4; figuring out how to add head tilt to imitation agents remains an open question.

## A.2  Hyperparameters

For APPO we use almost all the default hyperparameters of SampleFactory [29] at commit aed6cc92a7eb3510c4d4bcfac083ced07b5222f9. However, we use a batch size of 7168 and scale the observations by 10.0.

For Imitation Learning we use a learning rate of $3 \cdot 10^{-4}$, a batch size of 512, and stack a total of 5 states together to endow the agents with memory.

## A.3  State space Details

Since likely control architectures require the vector to be of a fixed size, we select a subset of visible road points and objects if there are too many and pad with a vector of $0.0$'s if there are too few. We sort both the road points, objects, and stop signs by distance and return the $500, 16, 4$ closest ones respectively where all three values are configurable by user. Using these values, the total state dimensionality is 6727. We note that this is a fairly high dimensional state and intentionally do not prune it to be smaller as dealing with this high dimensional input is a meaningful part of the benchmark. However, since we expect that users will likely want to construct their own types of states, utility functions are also provided to allow users to query the set of observed road points and objects.

## A.4  Features

The features of the ego object are:

- The speed of the object.
- The distance and angle to the goal.
- Its width and length.
- The relative speed and heading to the target speed and heading.

Road object features are:

- The speed of the object.
- The angle between the velocity vectors of the object and of the ego vehicle.
- Its width and length.
- The angle and distance to the road object's position relative to the ego vehicle.
- Its heading relative to the ego vehicle.
- A one-hot vector indicating whether the object is a vehicle, pedestrian, or cyclist.

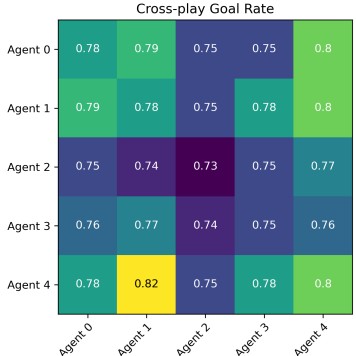
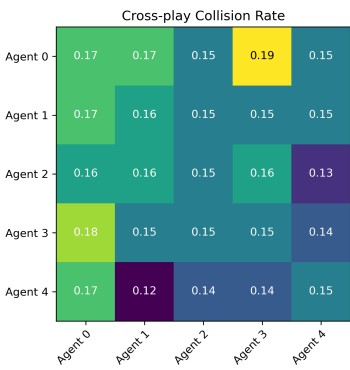

Figure 6: Goal rate (left) and collision rate (right) of agents when 50% are sampled from one side and 50% from another. The diagonal represents the baseline performance of the seed while element i, j represents seed i playing vs. seed j.

As per the VectorNet representation, each road point is part of a discretized polyline with an approximate discretization size of $0.5$ meters. To ensure that any vehicle is able to discern which road points are connected to each other, we include a vector pointing to the neighbor of the point in the polyline. As with the road object representation, all points are in the frame of the observing vehicle. Road point features are:

- The angle and distance to the road point's position relative to the ego vehicle.
- A 2D element representing the vector pointing from the current road-point to its neighbor in the polyline.
- A one-hot vector indicating whether the road point is a lane-center, a road line, a road-edge that can be collided with, a stop-sign, a crosswalk, a speed-bump, or is of unknown type.

## B Zero-shot coordination

We investigate whether the agents are learning incompatible conventions across seeds (ZSC) by taking the seeds from our best performing agent on the test set (the agent trained on all files) and sampling half the agents from one seed and half from another. We then perform this procedure across all 5 seeds, running each pair on the same set of 100 files. The results of this are summarized in Fig. 6 and appear to indicate that at the current level of agent performance there does not appear to be a zero-shot coordination issue. However, it is still possible that one might emerge as the agents become more capable and learn arbitrary symmetry-breaking conventions that are not compatible across agents. This argument lines up with the results of Sec. 5.3 which shows that the agents fail at high rates in interactive scenes; it may be necessary to get to lower failure rates in such settings for ZSC issues to emerge.

## C Ablation of the targets

The experiments described in the main body of the text provide all the agents with a target position, speed, and heading for their goal. In this section we investigate the effect on some of the metrics of relaxing the speed target requirement. Based on the results from Sec. 5.1, we only run experiments using 10000 training files as this appears to be sufficient to close the train-test gap and allows us to minimize the computational requirements of these ablations.

Table 3 lists a comparison between the original results and the ablation on target speed. Note that since the target requirements are relaxed, the target may become easier to achieve which in turn drives up the goal rate. However, we observe the following features. First, the collision rate increases sharply relative to APPO and the goal rate decreases markedly. It is possible that the speed target serves as a hint or curriculum that makes solving some of the scenes easier; for example some scenes could be solvable by simply linearly interpolating between the agent speed and the target speed. Since the requirements for reaching the goal are less constrained which allows for a greater range of

possible trajectories to reach the target, the ADE increases and the FDE increases significantly. The sharp increase in FDE likely occurs due to the agent being able to reach its goal at higher speed and since the post-goal behavior is out-of-distribution, the agent attempts often unsuccessfully to route itself back to its goal.

Table 3: Overview of metrics across methods for an 8 second rollout on the test set.

| Algorithm | Collision Rate (%) | Goal Rate (%) | ADE (m) | FDE (m) |
|---|---|---|---|---|
| Expert Playback | 4.9 | 100 | 0 | 0 |
| APPO | $20.3 \pm 0.8$ | $71.7 \pm 0.7$ | $3.1 \pm 0.2$ | $6.1 \pm 0.3$ |
| APPO w/o target speed | $29.3 \pm 1.2$ | $64.9 \pm 0.9$ | $4.3 \pm 0.3$ | $12.6 \pm 0.9$ |
| BC | $38.2 \pm .1$ | $25.3 \pm 0.1$ | $5.6 \pm 0.1$ | $9.2 \pm 0.1$ |

## D   Vehicle Model

Our vehicles are driven by a kinematic bicycle model [31] which uses the center of gravity as reference point. The dynamics are as follows. Here $(x_t, y_t)$ stands for the coordinate of the vehicle's position at time $t$, $\theta_t$ stands for the vehicle's heading at time $t$, $v_t$ stands for the vehicle's speed at time $t$, $a$ stands for the vehicle's acceleration and $\delta$ stands for the vehicle's steering angle. $L$ is the distance from the front axle to the rear axle (in this case, just the length of the car) and $l_r$ is the distance from the center of gravity to the rear axle. Here we assume $l_r = 0.5L$.

$$\dot{v} = a$$
$$\bar{v} = \text{clip}(v_t + 0.5 \, \dot{v} \, \Delta t, -v_{\max}, v_{\max})$$
$$\beta = \tan^{-1}\left(\frac{l_r \, \tan(\delta)}{L}\right)$$
$$\quad = \tan^{-1}(0.5 \, \tan(\delta))$$
$$\dot{x} = \bar{v} \, \cos(\theta + \beta)$$
$$\dot{y} = \bar{v} \, \sin(\theta + \beta)$$
$$\dot{\theta} = \frac{\bar{v} \, \cos(\beta) \, \tan(\delta)}{L}$$

We then step the dynamics as follows:

$$x_{t+1} = x_t + \dot{x} \, \Delta t$$
$$y_{t+1} = y_t + \dot{y} \, \Delta t$$
$$\theta_{t+1} = \theta_t + \dot{\theta} \, \Delta t$$
$$v_{t+1} = \text{clip}(v_t + \dot{v} \, \Delta t, -v_{\max}, v_{\max})$$

## E   Calculating Infeasible Goal Statistics

For computing the percentage of vehicles that must cross a road edge to get to their goal, we use the following procedure. We take the vehicles and shrink their width by 0.1 and their length by 0.3. These vehicles are very thin and so do not accidentally collide with a road edge due to small errors in their position. The scaling of the length ensures that the vehicles are not placed in a starting position where they have collided. Using this procedure we calculate the 3% statistic.

## F   Computing the simulator speed

We use two procedures for determining the overall speed of our simulator. The first measures the time to compute observations and step a single agent, which is what is reported as the simulator speed in

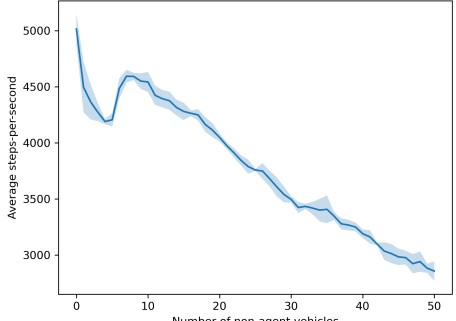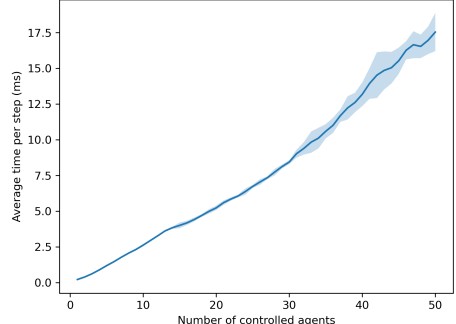

Figure 7: (Left) **Single-agent**: average steps-per-second as a function of the number of non-agent vehicles in the scene, where each step corresponds to computing observations for a single agent and stepping the simulation. (Right) **Multi-agent**: average time to compute observations and step all the controlled agents in a scene as a function of the number of agents in the scene. Shading corresponds to the standard deviation of the mean. The x-axis is cut at 50 as there are few scenes above this value.

the paper, and the second presents the time to compute observations and step a group of agents. In the first procedure, we randomly select a set of 2000 files and step through them from the first to the last time-step. At each time-step, we sample a random agent, set the remaining agents to be replayed via expert data, compute the observation for the sampled agent, compute a random action for the sampled agent, and then step the simulator forwards. We repeat this procedure 5 times and report average results and standard deviation of the mean values. Over this dataset and procedure, the single-agent steps-per-second (SPS) is around 3400. This is a slightly modified version of the procedure used to compute FPS in MetaDrive [22] to account for variations in the number of agents available in a scene as well as the lack of rule-based agents in our simulator. With 10 non-agent vehicles we achieve a SPS of 4543, 4049 at 20, and 3497 at 30. The left side of Fig. 7 shows the scaling with the number of other vehicles in the scene which includes the costs of computing partial observability, collision checking, and stepping the bicycle model.

We also perform a multi-agent variant of this procedure to test the scaling of our simulator with the number of controlled agents. Here we sample observations for all potential controlled agents in a scene and then take a simulator step. Again, we repeat this procedure 5 times, each time sweeping across all scenes in the 2000 files as defined in the prior procedure, and report average results and standard deviation of the mean values over the runs. The right side of Fig. 7 shows the scaling of the time to perform this procedure as a function of the number of controlled agents in the scene. We observe a seemingly linear scaling with agent number. There are about 8.5 agents on average and we achieve an average per-agent SPS of 452, allowing us to construct about 3800 frames of experience per-second.

## G   License Details and Accessibility

Our code is released under an MIT License. The Waymo Motion dataset is released under a Apache License 2.0. The code is available at `https://github.com/facebookresearch/nocturne`.