# OpenReview forum: "Nocturne: a scalable driving benchmark for bringing multi-agent learning one step closer to the real world"
_NeurIPS.cc/2022/Track/Datasets_and_Benchmarks — NeurIPS 2022 Datasets and Benchmarks _

### Official Review · Reviewer_9pmT · 2022-07-25
**A potentially useful multi-agent partially driving benchmark but needs more technical work.**

**Rating:** 6
**Confidence:** 2
**Correctness:** The benchmark appears to have appropr…
**Clarity:** Much of this paper is easy to understand

**Strengths:**

- Paper makes a good case for the importance of studying the driving benchmark. Compared to previous works, Nocturne has more efficient environment interaction efficiency.
- The paper describes the task, state action space design, and algorithm details. It is convenient for readers to reproduce the result and adopt this benchmark.

**Weaknesses:**

- The paper lacks the necessary algorithm comparison. The author claims it is a multi-agent benchmark, but the paper only includes PPO and BC.
- The paper mentions that the purpose of this benchmark is to study the multi-agent learning process in the real world, but does not clearly point out what advantages this paper has over previous simulators for this purpose.

**Additional Feedback:**

It is necessary for the authors to add more baselines to make this work more complete. The current experimental part is sparse.

**Documentation:**

Open source codebase is provided

**Ethics:**

I did not have any ethical concerns with the paper.

**Relation To Prior Work:**

The relationship to prior works is fairly clear. But I think the authors need to provide a more detailed comparison of advantages relative to previous work.

**Summary And Contributions:**

This paper introduces a driving benchmark. Relative to previous work, the authors claim Nocturne is the only available simulator that can compute an agent’s visible objects and step the agents dynamics at above 2000+ steps-per-second. The paper compares the performance of Expert Playback, APPO, and BC in different driving scenarios.

---

> ### Author Response · Authors · 2022-08-11
> **Clarifications on relationship to existing work**
>
> Thanks to the reviewers for identifying that speed is a strength of the simulator and that it is straightforward for users to adopt the simulator. We appreciate their review and answer the questions and weaknesses below:
>
> *The paper lacks the necessary algorithm comparison. The author claims it is a multi-agent benchmark, but the paper only includes PPO and BC.*
>
> **Response:** we appreciate the reviewer pointing this out. Our intent in this paper was primarily to introduce the simulator and the challenge, the goal and collision results are merely intended to provide a starting point for other researchers rather than a comparison of performance of different algorithms. Perhaps the reviewer would prefer that we use the word “challenge” instead of “benchmark” to better convey our intent?
> *The paper mentions that the purpose of this benchmark is to study the multi-agent learning process in the real world, but does not clearly point out what advantages this paper has over previous simulators for this purpose.*
>
> **Response:** We attempted to address this concern in both Table 1 and in the text of the related work section but from this concern it is clear that was insufficient. We summarize the distinction here and would deeply appreciate help from the reviewers as to where we failed to make clear how our simulator differs from others. Nocturne is, to our knowledge, the only simulator that can compute the set of partially observable objects and road points without rendering images. This is what allows our simulator to be fast and in turn makes it accessible for multi-agent learning. We have added a states-per-second column to Table 1 to make this clearer as well. In addition, the benchmark constructs a huge diversity of scenes by drawing on the waymo dataset, a feature that is not shared by all available benchmarks / simulators though is shared with MetaDrive and NuPlan (which is not yet fully released and currently consists of a github page and a blog post).

---

> > ### Comment · Reviewer_9pmT · 2022-08-25
> > **Response to author**
> >
> > The author's answer solved my confusion and I'm ready to improve my score.

---

### Official Review · Reviewer_cEjR · 2022-07-26
**An efficient driving simulator built on real data for partially observed and generalizable MARL research**

**Rating:** 5
**Confidence:** 4
**Correctness:** 9s real data are too short to get sou…
**Clarity:** It is well written.

**Strengths:**

1. Designing MARL methods under partially-observed real-world scenarios can be challenging and valuable. Nocturne provides a good starting point and provides adequate benchmark results.
2.  According to the benchmark result of SMARTS, increasing the number of agents will degrade the simulation efficiency. Also, generalization experiments usually take billions of steps for each training dataset. Therefore, the simulator can run up to 2000+ FPS, which is important in MARL generalization experiments.
3. The experiment results are sound and insightful.

**Weaknesses:**

1. The reward function encourages the agent to follow the expert trajectories, which are produced with the observation of all traffic participants. Therefore, I still suggest including pedestrians and cyclists in scenarios in the next version.
2. Waymo motion dataset has a 20s version where 20s trajectories are further divided into 9s fragments for motion prediction, see: https://waymo.com/open/data/motion/#overview Consider using the 20s version to tame the short trajectory problem.
3. I am not sure whether the maps in Waymo data contain overpass bridges or not. If so, please filter these maps, since Nocturne is a 2D simulator.
4. Experiments changing the ratio of controllable vehicles in the scene could be conducted in the future, i.e. 50 % replay from data, 50 % MARL agents. It would be interesting to discuss the similarity and differences between agents trained in heterogeneous and homogeneous populations.
5. For picking up trajectory intersection, I suggest using Time to Collision (TTC), which additionally considers the temporal intersection.

typo:
In Table. 1, the reference of VISTA simulator is incorrect.

**Additional Feedback:**

N/A

**Documentation:**

The documentation is complete

**Ethics:**

There is no ethical issue

**Relation To Prior Work:**

As far as I know, it covers all related works

**Summary And Contributions:**

Nocturne is a 2D-driving simulator constructed on real-world data and designed for partially observed MARL research. It first reconstructs maps and replay objects' trajectories contained in real-world datasets, such as the Waymo Motion dataset. After that, traffic vehicles are turned into controllable agents with partial observability, and actuated to arrive goal region according to a learnable policy. The policy will control vehicles with discrete actions and follow the dynamics of the bicycle model. Due to the efficient ray casting implemented by C++backend, Nocturne is efficient and can run up to 2000+ FPS with rasterized image observation or vectorized observation. In the experiment, the authors conduct generalization experiments on the scenarios imported from the Waymo Motion dataset with RL and IL methods. The experiment shows that increasing the number of scenarios contained in training set will improve the test performance on the unseen holdout set. Also, the failure mode analysis and ZSC test indicate that improvement on current algorithms is required to solve the partially-observed coordination problem.

---

> ### Author Response · Authors · 2022-08-11
> **Response to questions**
>
> Thank you for the review and analysis of our paper! We appreciate your pointing out the benefits and hope that the responses below address concerns about the paper. We also appreciate your catching the incorrect VISTA reference. We have addressed some of these comments in the paper (marked in red) but also address each point individuall below.
>
> *The reward function encourages the agent to follow the expert trajectories, which are produced with the observation of all traffic participants. Therefore, I still suggest including pedestrians and cyclists in scenarios in the next version.*
>
> **Response:** We agree, on the Github page project page cyclists and pedestrians are part of the roadmap for the next release which can be seen here: [github project link](https://github.com/facebookresearch/nocturne/projects/1).
>
> *Waymo motion dataset has a 20s version where 20s trajectories are further divided into 9s fragments for motion prediction, see: https://waymo.com/open/data/motion/#overview Consider using the 20s version to tame the short trajectory problem.*
>
> **Response:** Unfortunately, while the recorded snippets are 20 seconds long on Waymo’s side, the released trajectories do not contain the 20 second snippets and it is not possible from the available information to string the 9 second snippets together into the full snippets from which they were sampled. If this were possible, we would absolutely do it.
>
> *I am not sure whether the maps in Waymo data contain overpass bridges or not. If so, please filter these maps, since Nocturne is a 2D simulator.*
>
> **Response:** This is a great point. While we have only ever found one example of an overpass in the data, this is worth being careful about. In the current benchmark version, if a scene contains an overpass and a vehicle is forced to interact with it, for example, if its goal is on the other side of the overpass, it will by default be labeled as an expert and replayed via expert data. This point is now added to the main text in the section on “oddities of the benchmark”.
> However, we would like to simply filter out all overpasses. We will shortly release an updated dataset that has all overpass bridges filtered out using the following method (you can see this on the project roadmap).. We go through all the vehicles in a scene and compute the minimum and maximum z coordinate. If this exceeds 3 meters (the average height of an overpass) we remove the example from the dataset. This brings the size of the dataset down to X from Y. Note that this method is likely just removing scenarios that contain hills but we consider this an acceptable trade-off to ensure that there are no overpass scenes included.
>
> *Experiments changing the ratio of controllable vehicles in the scene could be conducted in the future, i.e. 50 % replay from data, 50 % MARL agents. It would be interesting to discuss the similarity and differences between agents trained in heterogeneous and homogeneous populations.*
>
> **Response:**
> We agree! While due to computational constraints we are not able to run this experiment for this rebuttal, we intend to do so as soon as we are able.
>
> *For picking up trajectory intersection, I suggest using Time to Collision (TTC), which additionally considers the temporal intersection.*
>
> **Response:** We agree that this would be a good metric! Unfortunately this first release does not include lane-level information that would be required to figure out which vehicles can actually collide and makes calculating TTC very challenging. Unfortunately, the lane assignment data in Waymo Motion is still somewhat inscrutable to us and will require further discussion with the Waymo team to ensure that this is implemented correctly.

---

> > ### Comment · Reviewer_cEjR · 2022-08-25
> > **Experiment should be conducted on Waymo's 20s motion data**
> >
> > Thanks for your response.
> >
> > The released trajectories **do** contain the 20-second snippets. It could be found in the Waymo's google cloud with a path: *waymo_open_dataset_motion_v_1_0_0/uncompressed/scenario/training_20s*. I checked the trajectories in 9s motion data recently. I think 9s is too short to get a solid experimental result. In 9s cases, the displacement distances of all vehicles are short, even zero meters for some agents, making the scenarios easy to solve. Thus it is difficult to tell if the agents learned to simply move forward, or interact for solving the scenarios. Furthermore, 9s snippets contain insufficient interactions for MARL research where complex behaviors between agents should be investigated. Therefore, I think the experiments should be conducted on 20s Waymo cases instead of 9s cases.

---

> > > ### Author Response · Authors · 2022-08-26
> > > **Error in missing 20s dataset in response but 9s version is still standard / interesting**
> > >
> > > Our apologies for missing the 20 second dataset in our response. When we were designing the benchmark a while back, we decided not to use that dataset as there is not a corresponding validation and test set as well as the 9 second dataset being standard in the supervised learning setting. We then failed to remember the existence of the 20 second version when composing our response. However, we agree a 20 second version of the benchmark would be interesting and will add it to the timeline for the next release, split into a corresponding train, validation, and test set.
> > >
> > > However, we do not believe that the 9 second version is too short to get a compelling experimental result. First, the widely studied Waymo Motion prediction challenge itself is defined over 9 seconds, indicating that the community finds this variant challenging. Although we do a control task rather than a prediction task, if the prediction variant is challenging it is likely the case that the control task is challenging too (and that our methods do not solve it is evidence in that direction). Second, on our website [Nocturne Website](https://www.nathanlct.com/research/nocturne) we have added videos of clearly challenging 9 second tasks such as left turns, determining order of who should go, and highly obscured parking lots. Finally, although some vehicles do not need to move to achieve their goal, our definition of controlled agents only include agents that have to move some non-trivial distance to achieve their goal so these simple "stay in place" tasks are not included in the benchmark.

---

> > > > ### Comment · Reviewer_cEjR · 2022-08-28
> > > > **Response to the authors**
> > > >
> > > > Thanks for the further explanation.
> > > >
> > > > Actually, the reason why Waymo split the 20s record into several 9s slices is that autonomous driving systems usually predict future 8s according to the information in the last 1 second. This is only a standard forecasting problem defined by the industry. It **doesn't** mean that 9s are challenging enough and the community **can not** do 21s prediction. Also, though the videos are pretty interesting and reasonable, these visualized results are only a small part of recorded cases and can be carefully picked. They can not convince me that each case in the dataset is of the same quality. It is correct that some 9s scenarios are challenging, while the 20s dataset definitely contains more high-quality scenarios.
> > > >
> > > > I also checked the discussions with other reviewers. **My main concern is the quality of data**, which is also mentioned by *Reviewer #HwJ5*. This simulator is built on real data and thus the quality of the data is essential. In addition to the data collection's ego-centric nature, the length of the replayed episodes is also a key factor for interactive multi-agent autonomous driving. A long episode will definitely enable more complex and meaningful interaction.
> > > >
> > > > I really appreciate the huge effort to build the efficient simulation engine, while results on the 20s Waymo motion data should be included before considering publishing. By the way, I am not sure if there are some datasets collecting the traffic flow from a God's Perspective like a bird-eye camera at the intersection, which can prevent the occultation/partial observation problem due to the ego-centric data collection. If so, using these datasets can avoid several data-quality problems mentioned in the rebuttal period.

---

> > > > > ### Author Response · Authors · 2022-08-29
> > > > > **Response to reviewer**
> > > > >
> > > > > We appreciate the reviewer's feedback. However, we continue to think that 9 second interactions are challenging and sufficient to probe certain types of interactivity such as complex yields, merges, and heavily partially observed scenarios as evidenced by the posted videos. Given the existence of these interactions and that a set of tuned baselines failed to solve the tasks, there is clear value in continuing to study the 9 second version before jumping to the 20 second version. Therefore, we do not agree that “9 seconds is too short to get sound results.”
> > > > >
> > > > > As for potential benefits of adding 20s, we argue that a significant fraction of them are captured in the 9s version. One benefit of the 20 second segments is that they approximately double the number of interactions in an episode, but each individual interaction would still be challenging and be contained in the 9 second variant. While we agree that on 20 second timescales there will necessarily be more interactive events per episode and that some interactions can occur on longer time-scales, a lot can already happen in 9 seconds that is non-trivial and interesting.. We also note that the 20s files are the same format as the 9s file and we have documented scripts used to build the dataset, so users may easily build and use the 20s dataset if they wish to do so.
> > > > >
> > > > > As for the point about potential cherry-picking of the videos, these scenes were selected from a subset of 3x as many scenes to find scenes with distinct agent failure modes (i.e. we have 10 scenes on the website and selected them from 30 randomly drawn scenes; in total there are around 100k scenes we can draw from). Note that the 9s version of the dataset is purposefully pruned for interactivity on Waymo's end using conditional independence of the joint distribution as a proxy interactivity measure to filter on and should be expected to contain non-trivial prediction problems (which we did observe).Therefore, we do expect a significant fraction of the scenes to be “interesting” as evidenced in the videos.
> > > > >
> > > > > Finally, while there are bird-eye-view datasets such as HighD or INTERACTION, they are significantly smaller in both number of scenes and variety of settings from which they are sampled as they are often taken from stationary drones or fixed overhead cameras at intersections. A birds eye view would be great but at the moment would require trade-offs in the diversity of available scenes. Having a huge number of scenes, of which a significant fraction are complex, ensures really thorough testing of a proposed policy and is a key component of the benchmark that we think is worth keeping.

---

### Official Review · Reviewer_XS9C · 2022-07-27
**Interesting approach to get driving agents in a more realistic frame**

**Rating:** 8
**Confidence:** 2

**Strengths:**

* The simulator is run on real-world data at a high-frequency rate, providing an accurate account of driving situations.

* It combines coordination and cooperation, where the agents’ parameters are tuned to match experts’ capabilities.

* The work probes further into the ability of baseline RL multi-agents to handle complex scenes and provides an overview of their limitations, mainly during cooperation tasks.


**Weaknesses:**

* Given the simulator builds a 2D birds-eye view of the scene, it is inherently limited when datasets built from cameras of driving cars (such as WayMo) are employed, as, for instance, pedestrians had to be excluded, even though they represent one of the main challenges to tackle in autonomous driving.

**Additional Feedback:**

I find it intriguing that in Fig. 4 the average and final displacement error are both smaller for the test split than the train split, whereas the opposite is true for the % of goals achieved and collisions. Why is that? What does this tell us about these metrics? Have you tried drawing some of the paths taken by the agents and the experts to get a better notion of what is going on (maybe include them in the sup. material)?

**Clarity:**

The paper is well-written and easy to follow, with clear details of the limitations and contributions of the present work.

**Correctness:**

Both the experiment design and evaluation methods seem appropriate and correctly performed; the conclusions drawn are reasonable.

**Documentation:**

The Github repository provided is very clear, with all the necessary instructions to reproduce the experiments.

**Ethics:**

There are no ethical concerns in the present work.

**Relation To Prior Work:**

The authors are precise in how their contribution departs from previous efforts, providing a simple table which summarizes the differences.

**Summary And Contributions:**

The authors introduce a new 2D driving simulator, with a focus on multi-agent coordination under partial observability (hence the name ‘Nocturne’). Crucially, unlike previously published driving simulators, Nocturne makes use of state-based partial observability, computed through efficient intersection methods, disposing of the need of rendering camera images to acquire the set of visible objects in a given time. Thus, the simulator is able to run at over 2000 steps-per-second using real-world data.

---

> ### Author Response · Authors · 2022-08-10
> **Review appreciated, clarification on lack of pedestrians**
>
> **Response:** We thank the reviewer for their comments and their kind words about the paper.
>
> For the mentioned weakness on the absence of pedestrians, we want to clarify that the birds eye view does not present a challenge for pedestrians and that a version of the benchmark could straightforwardly be released containing pedestrians. Pedestrians were not used in this version as we require a different dynamics model for the pedestrians i.e. it would not be correct to use a bicycle model for the pedestrians. Since we wanted to ensure that we had a good pedestrian dynamics model, we deferred the inclusion of pedestrians to future work.
>
> As for the question about understanding the train test split for ADE and FDE, we have attempted to draw the trajectories but have not yet found a visualization that is informative of the train-test split. However, given the overlap of the std. error, it’s possible that the train-test split is illusory. We have, however, added a wide variety of videos to the webpage, split by train, test, and our interactivity metric, that are hopefully informative.

---

### Official Review · Reviewer_HwJ5 · 2022-07-27
**A fast partially observable simulation framework for self-driving datasets**

**Rating:** 6
**Confidence:** 4
**Correctness:** The claims of the paper seem sound an…

**Strengths:**

- Nocturne is highly performant ... 2000+ steps/second makes this a feasible environment for RL learning.

- Nocturne supports both vectorized and rasterized representations, this allows for a greater diversity of approaches to be developed.

- The Nocture benchmark is built upon a well-respected and utilized self-driving dataset.

**Weaknesses:**

- The visibility map seems pessimistic.  It precludes the ability of a driver to see over the hood of a neighboring car (for example).  This is not discussed and constitutes a real weakness of the fast method introduced here in comparison with those generated from imagery or lidar.

- Given the source of the data, there are regions of space that are visible to actor "A" but were occluded from the Waymo vehicle that collected the data.  There does not appear to be any way to represent potentially missing data within Nocturne.

- I would like to see a more compelling argument for the value added by a dynamic visibility query.  It seems a reasonable alternative would be to precompute a visibility map as a function of time given the expert trajectory.  Clearly, these could diverge, but do they?  Are the differences between a dynamic vs. a static visibility map significant in this context?

- At this point, BC seems to be quite a weak baseline.  There are a plethora of goal-conditioned forecasting methods which would be more appropriate here.

- Intersections as a proxy for interactions seem to be a weak proxy.  Vehicles that merge into the same lane 8s apart will be considered interacting ... while two vehicles performing complementary left turns will not.

- The Zero Shot learning section seems like it should be supported by a simple statistical test.  (i.e. there is no significant difference between self-play and cross-play).

- I would really like to see some discussion or perhaps ablation experiments w.r.t. to the conditioning information provided.  i.e. what happens to performance if we do not provide the final velocity?

**Additional Feedback:**

- Much of the motivation for the manuscript is about getting the performance necessary for RL in partially observable environments.  The manuscript would benefit from additional motivation for why we should care about multiagent planning in a self-driving setting?

- It appears that most failures come from collisions, however, there is a slice of failures without a collision.  It would be nice to see a deeper discussion of these examples.

- The work would also benefit from videos on the website of the learned agents.

**Clarity:**

- [nit] "road actor" or "agent" is more in line with other papers than "road object" for potentially dynamic actors.

- "discretized at a rate of 0.1 Hz with" should be 10Hz or 0.1s I believe

- In figure 3 ... please label the x axis with the values ... i.e. 10, 100, 10000 etc (rather than 1, 2, 3, 4, 5)

- The start of section 4 claims "We run several learning methods" ... I would make a more modest claim.  "We run both a standard RL and IL baseline ..."

- It is not clear to me why actors should be "removed" upon colliding with a road edge?  Don't they continue to provide important social context?

- [nit] I would commit to "APPO" or "PPO" rather than using the two interchangeably.

- Figure 5 does not have any logarithmic scale (despite the caption).

- The appendix indicates that a time cutoff is required to avoid a trivial "creeping" solution.  That doesn't seem to be a problem given that there is a speed requirement on achieving the goal.

- The benchmark is explained much more clearly in the appendix than in the main paper.  Given the null result of zero-shot coordination, one could consider moving the figure to the appendix to make space for a more detailed description of the benchmark task in the main paper.

**Documentation:**

- The documentation seems fine on the external website and in the supplementary material, though the link to a blog post is non-functional.
- The vehicle model section would benefit from a cartoon diagram.

**Ethics:**

I have no ethical concerns with this work.

**Relation To Prior Work:**

- Given the centrality of "speed" to Nocturn's value proposition, I would like to see estimates of steps/second in the table 1 comparison.

- This work would benefit from additional discussion of the Waymo Open Motion dataset and/or other relevant self-driving datasets.



**Summary And Contributions:**

Nocturne introduces a 2D simulation environment to support RL/IL approaches to multiagent planning in the context of autonomous driving.  Nocturne seeks to improve on other work in two key ways.  First, Nocturne is able to generate 2D views of the world that are visible to any actor.  Second, Nocturne is fast ... allowing for 2000+ steps/second.  Speed is absolutely essential due to the sample inefficiency of RL algorithms.  Nocture achieves these fast visibility queries by leveraging techniques from the computer graphics community.

While Nocturne can be applied to multiple datasets, the authors introduce a benchmark based on scenarios from the Waymo Open Motion dataset.  The Nocturne dataset includes all scenarios which do not interact with a traffic light (134k/487k scenarios).

The authors then train RL and IL agents on the benchmark, investigating the performance of the agents as a function of the number of training scenarios.

---

> ### Author Response · Authors · 2022-08-10
> **Extremely thorough review, thanks! Clarifying answers and updated manuscript attached.**
>
> Clarity issues: We thank the reviewer for catching some mistakes, thoroughly reviewing the appendix, and so extensively reviewing the paper in a way that will make it a better paper! The issues that can be addressed without discussion are in the updated manuscript and marked in red. In particular, APPO is consistently used, “several learning algorithms” has been replaced with the suggested text, and the logarithmic scales have been fixed. With the additional page we have taken advantage of it to move the rules and explanations of the benchmark out of the appendix and into the main paper, replacing the cross-play results for space. We believe the updated paper addresses the majority of the indicated issues.
>
> As for the remaining questions:
>
> *[nit]: "road actor" or "agent" is more in line with other papers than "road object" for potentially dynamic actors.*
>
> **Response:** We use road object to remain consistent with the naming of vehicles, pedestrians, and cyclists in the Waymo Motion paper and codebase. We think this is a sensible choice to avoid confusing users of the dataset but if the reviewer feels strongly that it should be changed, we are happy to do so.
>
> *Question: It is not clear to me why actors should be "removed" upon colliding with a road edge? Don't they continue to provide important social context?*
>
> **Response:** We agree that it would also be possible to keep the vehicles in the scene instead; we believe both positions (keep vs. not keep) to be defensible. However, there are two motivations behind removing. First, at equilibrium we expect the collisions with road edges to be 0 i.e. the optimal policy should never collide with a road edge, so any set of rules over removal / non-removal of a vehicle when colliding road edges should be fine as long as colliding with a road edge is always suboptimal. Second, we wanted to avoid potential cases where a vehicle might temporarily collide with a road edge to somehow simplify the path to its goal; this could be avoided by adding a penalty to the reward but we did not want to force users to include a penalty term as we want reward shaping to be a researcher’s decision.
>
> *The appendix indicates that a time cutoff is required to avoid a trivial "creeping" solution. That doesn't seem to be a problem given that there is a speed requirement on achieving the goal.*
>
> **Response:** The speed requirement is only a final-speed requirement, there are cases where the speed requirement can be satisfied by creeping for a while and then speeding up right at the end. Furthermore, we know that the tasks can all be accomplished within this time-frame because of the expert data so it seems reasonable to insist that this occurs. However, we agree that other ways of removing the creeping solution are possible.
>
> *This work would benefit from additional discussion of the Waymo Open Motion dataset and/or other relevant self-driving datasets.*
>
> **Response:** Thanks for the suggestion. We agree with this and have added a short datasets section to the related work.
>
> *Given the centrality of "speed" to Nocturn's value proposition, I would like to see estimates of steps/second in the table 1 comparison.*
>
> **Response:** This is a good suggestion but something we initially avoided as
> Response: This is a good suggestion but something we have avoided as
> - it can be challenging to ensure that we have maximized the performance of all the other libraries on our hardware and wanted to ensure a fair comparison. We would also be forced to compare across quite different sets of scenarios as no two simulators support the same set of scenario configurations.
> - it only seemed correct to compare to simulators that support partial observability. Every other simulator we are aware of does this by rendering images and it seemed fairly clear that this would be several times slower than computing partial observability via computational geometry methods. The fastest example we are aware of, MetaDrive, computes somewhat imprecise images at 300 FPS according to their own paper, though, we note that this calculation includes 10 rule based agents in the scene (whose speed to step we assume to be negligible as is the case for simple rule based agents). CARLA operates at around 60 FPS: [CARLA 0.8.0 notes](http://carla.org/2018/03/27/release-0.8.0/)
>
> We have included speed values in the table for the partially observable simulators  where the authors provided values in their paper or codebase as we assume the authors are attempting to maximize performance on their hardware.
>
> *The work would also benefit from videos on the website of the learned agents.*
>
> **Response:** We have added them to the website now for a wide variety of scenes / found failure modes of the agents. Thank you for the suggestion.

---

> > ### Author Response · Authors · 2022-08-10
> > **Response part 2**
> >
> > *The visibility map seems pessimistic. It precludes the ability of a driver to see over the hood of a neighboring car (for example). This is not discussed and constitutes a real weakness of the fast method introduced here in comparison with those generated from imagery or lidar.*
> >
> > **Response:** This is a good point and a discussion is now added to the paper in the section on our partial observability model. However, we note that we can construct modifications to the shape of the car that make it a convex body with a hood region and still quickly construct the visible region without having the view blocked by the hood portion. We can easily add this as a feature in future releases as we push the benchmark towards increasing realism and intend to do so. However, the reviewer is correct that including a full 3-D model of partial observability would require a completely different set of computational techniques (for example, to account for the fact that very tall cars can see over quite small cars).
> >
> > *Given the source of the data, there are regions of space that are visible to actor "A" but were occluded from the Waymo vehicle that collected the data. There does not appear to be any way to represent potentially missing data within Nocturne.*
> >
> > **Response:** If we understand the reviewer correctly, they are raising the point that due to the egocentricity of the data collection process, there are vehicles or other road agents that are not persistently visible or may never be visible to the Waymo vehicle. This is true and an unavoidable consequence of working with egocentric data though we consider the benefits of a data-driven, diverse benchmark to outweigh these downsides and that it may be possible to infer the missing vehicles or insert them using generative techniques. However, we note that even if some objects are not observable, unobservable objects only influence why an agent might have taken the trajectory it took, not whether the goal is itself achievable. Hence, all the goals we are requesting of our agents are valid and challenging even if it might be challenging to understand why a trajectory was taken.
> >
> > *I would like to see a more compelling argument for the value added by a dynamic visibility query. It seems a reasonable alternative would be to precompute a visibility map as a function of time given the expert trajectory. Clearly, these could diverge, but do they? Are the differences between a dynamic vs. a static visibility map significant in this context?*
> >
> > **Response:** Static visibility maps pose several challenges. First of all, they would make it difficult for us to include the head rotation as a possible action, a feature we consider essential to our simulator for modeling how humans acquire information given their limited field of view. Second, unless the agents follow directly along the path of the expert, the static map would quickly become wrong, which would occur frequently at the beginning of learning for RL agents. For many scenes, there are likely multiple possible ways to solve it but using a static map would essentially make it necessary to follow the expert trajectory and preclude potential diverse solutions. Third, without dynamic occlusion it is impossible to construct new scenarios that do not have expert data. Finally, we are interested in the use of Nocturne as a tool for studying the impact of occlusion (and potentially distraction) on emergent coordination and it seems essential to have dynamic inclusion for this. If the reviewer considers this argument to be essential to the paper, we are happy to include it.
> >
> > *At this point, BC seems to be quite a weak baseline. There are a plethora of goal-conditioned forecasting methods which would be more appropriate here.*
> >
> > **Response:** We agree that there are goal-conditioned forecasting methods could be found that outperform BC. Our sole intent in including BC as a baseline was to demonstrate that it was possible to reproduce a simple, standard result in our environment and note that the results closely match the BC baselines in “Symphony: Learning Realistic and Diverse Agents for Autonomous Driving Simulation”

---

> > > ### Author Response · Authors · 2022-08-10
> > > **Response Part 3**
> > >
> > > *Intersections as a proxy for interactions seem to be a weak proxy. Vehicles that merge into the same lane 8s apart will be considered interacting ... while two vehicles performing complementary left turns will not.*
> > >
> > > **Response:** We have added a mention to this paper of the limitation of this interaction metric and on the plots intentionally use the axis “number of intersecting paths” rather than “number of interactions” to avoid misleading the reader as to the strength of this proxy. While there are certainly missing interactions that our metric does not capture, it is still highly predictive of agent failure and thus is useful for highlighting scenarios that agents fail on (we have added examples on the website of agent behaviors in scenes of different interactivity). It definitely does not capture all interactions but the interaction metric used in the original Waymo motion paper, measuring the conditional independence of the trajectories, is difficult to estimate given the recurrent policies used here. Finding tractable interaction metrics is, we believe, still an open problem in the field.
> > >
> > > *It appears that most failures come from collisions, however, there is a slice of failures without a collision. It would be nice to see a deeper discussion of these examples.*
> > >
> > > **Response:** these are just agents that have missed or fallen short of their goal; the targets are quite tight and it is possible that the discretized action space used here can make it challenging to get the appropriate tolerance. This should hopefully visible from the videos that we uploaded to the website.
> > >
> > > *I would really like to see some discussion or perhaps ablation experiments w.r.t. to the conditioning information provided. i.e. what happens to performance if we do not provide the final velocity?*
> > >
> > > **Response:** we are working on running this experiment and will report results when they are done. However, we figured it was worthwhile to start a discussion on the above questions and issues in the meantime. However, as clarification, presumably in this experiment the reviewer does not intend for there to be a velocity target as final velocity information is needed if there is a target velocity for achieving the goal.

---

> > > > ### Author Response · Authors · 2022-08-16
> > > > **Initial ablation results**
> > > >
> > > > We have uploaded the results of the speed ablation to the appendix. We observe that since the final speed target is less constrained the policies begin to diverge significantly more from the expert trajectories. Surprisingly, however, we also observe an increase in the collision rate and a decrease in the goal rate. We propose an explanation for these results in the appendix (namely that decreasing the constraints on the goal makes it so that more trajectories can feasibly reach the goal and do so faster) but note that further study would be required to verify the hypothesis proposed there.

---

> > > > ### Comment · Reviewer_HwJ5 · 2022-08-16
> > > > **Part 3 response**
> > > >
> > > > Thank you, the ablation results are an important addition.

---

> > > ### Comment · Reviewer_HwJ5 · 2022-08-16
> > > **Part 2 response**
> > >
> > > Again, thank you for the comprehensive response to my original review.  The authors have resolved most of my questions, but I would like to continue discussing the data collection's ego-centric nature.
> > >
> > > I agree with the authors on the following - "This is true and an unavoidable consequence of working with egocentric data though we consider the benefits of a data-driven, diverse benchmark to outweigh these downsides"
> > >
> > > My suggestion, however, is that this is accessible in the data itself.  I.e. I should have access to a mask that indicates whether or not a region of space was visible during the data collection process.  A policy may want to respond differently when planning through space visible to the agent (per nocturn) but essentially empty values (given data collection).

---

> > > > ### Author Response · Authors · 2022-08-17
> > > > **Response to masking suggestion**
> > > >
> > > > We understand the question better now. As a quick clarification in case this is what is being referred to, information about what is partially observed is implicitly available in the vectorized state i.e. if the state contains a car with a certain width and heading, then there is nothing visible behind it, and explicitly available in the image-based state though learning from images is not done in this paper.
> > > >
> > > > However, we think that the reviewer is referring to what state was historically partially observed during the egocentric data collection process to give the agent a sense of where historical data might be missing; this is an interesting suggestion! We have two responses to this: 1) this is challenging due to missing information in the data and 2) in case the reviewers concern is that this information is necessary or helpful to actually solve the task, Nocturne scenes are constructed so that is not. We elaborate on these two explanations below.
> > > >
> > > > With respect to computing which areas were not properly observed during data collection, there are two key issues. First, it is not clear which car in the scene is the data collecting car as this information appears to be left out of the dataset (we suspect but have not confirmed that this data is not available as it would allow users to do imitation learning on the Waymo vehicle). We could try to infer which vehicle is the ego vehicle by reasoning about exactly which objects are disappearing but it would be difficult to verify if this method is working correctly without ground truth information. As an aside, the challenge is to construct agents that can operate in generic scenes where this mask would not be available as it is a form of historical data about a particular scene as well as implicitly encodes trajectory information about an expert that would also not be available.
> > > >
> > > > As for a potential concern related to the necessity of this data for solving the scene, scene construction in Nocturne is specifically set up to mitigate the challenges of the partially observed data collection and ensure that the planning problem posed to the agents has a solution that would not require this historical information. While the goals each agent is attempting to achieve are occasionally strange due to egocentric data collection, each goal provided to the agent is achievable given the local information despite the fact that some road objects that were in the real world scene are missing. This would not, for example, be the case if instead of only including vehicles that were available at scene initialization we dynamically added in vehicles that were available at later times. Vehicles would occasionally collide with a vehicle that had been dynamically inserted and agents would somehow have to account for the fact that a vehicle could suddenly appear anywhere.

---

> > > > > ### Comment · Reviewer_HwJ5 · 2022-08-27
> > > > > **Continued discussion of masking**
> > > > >
> > > > > With the caveat that I understand the limitations of not having access to the ego vehicle in Waymo Open (it is available in other datasets, such as Argoverse 2), I want to provide an illustrative example of why I think this matters.  Perhaps within the context of this example, the authors can better illustrate why this is unimportant.
> > > > >
> > > > > Assume we have a T-intersection.  The ego vehicle is at a stop sign at the terminating leg of the T.  There is a building to its right ... so the area to the right of the ego vehicle is largely occluded.
> > > > >
> > > > > Now we have another actor (the actor of interest here) who is about to traverse the T intersection via the non-terminating portion of the T.  The future behavior of our new actor is strongly dependent on whether or not there is a stopped vehicle in the roadway ahead (which should be visible to our actor), but is missing because the area is occluded from the ego vehicle.
> > > > >
> > > > > This information (whether or not the road ahead is blocked) seems vitally important to the planning problem.   It seems important that an agent at least knows that the information is missing (allowing to reason about that source of uncertainty).

---

> > > > > > ### Author Response · Authors · 2022-08-29
> > > > > > **Thanks and clarifying example**
> > > > > >
> > > > > > Thank you very much for the thorough review process, taking the time to engage with us, and detailed feedback. We appreciate your willingness to increase your score.
> > > > > >
> > > > > > As for the example posted, we are still trying to make sure we understand but let us describe our understanding and see if it is correct. We've constructed the following diagram that we think illustrates the described scene: [Google slides link](https://docs.google.com/presentation/d/1kxposDjZ2u8DMvvh7P0MasFYP8gVpcnbpGqBJ28KGDg/edit?usp=sharing). The first slide illustrates the scene during data collection, the second the corresponding Nocturne scene that is constructed from the data. In case this is the source of disagreement, **note that the only agents in the scene are those that were available at time t=0**. If we instead inserted vehicles at later times in the episode when they became available in the data, there would be cases where a vehicle would be inserted directly on top of another vehicle.
> > > > > >
> > > > > > Now we describe our understanding of the example to see if we got it right. In this setting, driving agent A will have some final position in the data that is either short of obscured agent O or near O's position (if O moved during the collection process). However, if ego agent E never observes O, then the nocturne scene will simply not contain O and agent A will have an easier task than it would have had in reality, namely, driving towards a T junction while making sure not to collide with stopped vehicle E. The fact that there was an obscured vehicle there at collection time affects the position of the goal, but does not require driving agent A to know about the obscurity at collection time as it can clearly see that there is no vehicle in the right side of the T in the corresponding Nocturne scene. It is possible that we have misunderstood this example but would really like to understand if the reviewer is able to explain more. Adding support for Argoverse 2 is on our next release timeline and would include this obscurity during collection time mask if it would be helpful.

---

> > > > > > > ### Comment · Reviewer_HwJ5 · 2022-08-29
> > > > > > > **continued discussion**
> > > > > > >
> > > > > > > Your slides perfectly capture the scenario.  Thank you.  And the comment about Argoverse was an FYI (not feedback for this manuscript).
> > > > > > >
> > > > > > > I am working under the assumption that we are trying to build a model of the agent's behavior/policy.  In the scene we have drawn here, the goal itself seems arbitrary and random.  It only makes sense in the context of their being an actor blocking the lane ahead.  That is, we do not want a policy to learn: (I have ROW, the road ahead is clear, and I will decel and stop).  However, if the agent had access to information about occlusions in the data collection process, the goal makes a lot more sense (I have ROW, while I can see the road ahead, I have no information as that data was not collected, ... I might need to decel and stop).
> > > > > > >
> > > > > > > Contrary to your statement about "having an easier task", I am concerned that without any information about missing values, you are constructing a very difficult task for the proposed policy (i.e. you are setting up situations where the goal is unpredictable given the information available).  I understand the goal is given, but it seems to me this understanding is still important when attempting to learn a planner.

---

> > > > > > > > ### Author Response · Authors · 2022-08-29
> > > > > > > > **continued discussion**
> > > > > > > >
> > > > > > > > Ah, okay, I think we are finally clear; thank you for working this through with us! Based on this, our understanding is that the underlying issue is the oddity of some of the goals and their occasional incomprehensibility without information about the obscurity map of the ego vehicle. So, we think that there is both validity to this critique and a set of ameliorating factors that make this less of an issue.
> > > > > > > >
> > > > > > > > As an example of ameliorating factors, we think this particular example actually still makes sense to the agents despite the obscurity. From the perspective of the agent with ROW, while it may be unclear to the agent **why** it has this goal since the right portion of the T is free, the goal itself is achievable. From the perspective of the ego agent, since it cannot see around the corner, it also needs to contend with the possibility that there is an obscured car and that the driving agent might come to a stop in the middle of its path. The key distinction is that we are not exactly trying to build a model of the agent's behavior / policy but generate a goal conditioned agent with whatever goals are taken from the data; the agent does not need to infer its own goal but contend with whatever goals we have provided.
> > > > > > > >
> > > > > > > > However, we absolutely agree that there are scenarios where the goals will not be predictable for other agents in the scene and that this creates challenges. Consider a scene where vehicles are driving towards stopped traffic but the stopped vehicles were not observed by the ego vehicle that collected the data. When constructing the corresponding Nocturne scene, one of the agents has a goal of coming to a stop on the highway, but the other agents cannot infer this goal since to them the road ahead is totally unobstructed. While providing the obscurity mask as an input would be a partial resolution to this issue, it would lead to policies that cannot be used as a human driving model as at deployment time in new scenes the obscurity mask will not be available as an input. As an interesting alternative, we could use the obscurity mask as a means of inserting new vehicles into the system that generate "boundary conditions" that are consistent with the behavior of the agents in the scene e.g. in your example we would infer from the collected data that there was a vehicle on the right arm of the T and include it in the scene. While it's not the original proposal, we think this is an extremely compelling idea and will definitely look to include the obscurity mask as an available feature when we work on including Argoverse 2. Thank you for the discussion!

---

> > ### Comment · Reviewer_HwJ5 · 2022-08-16
> > **Part 1 response**
> >
> > Thank you very much.  All of the part 1 comments are well addressed and I believe the work is better for the changes.  One exception is the videos ... where can these be found?  I cannot seem to find them on the github page.  I also cannot find another website.
> >
> > Additionally, please check the formatting of your references.  For example, [4] seems to have unnecessary capitalization.

---

> > > ### Author Response · Authors · 2022-08-17
> > > **Thanks and link added**
> > >
> > > The videos can be seen at the [Nocturne Website](https://www.nathanlct.com/research/nocturne). The link can be found in the caption of Figure 2. Now that we have added videos of RL replays, we also link to the webpage in the results section, as well as in the Github README for visibility. Thanks for pointing this out!
> > >
> > > We have fixed reference [4] and have made sure the formatting of our references is uniform. We thank the reviewer for catching this!

---

> > ### Comment · Reviewer_HwJ5 · 2022-08-27
> > **Ready to improve score**
> >
> > I want to thank the authors for an extremely responsive discussion.  I am ready to improve my score.

---

### Meta-Review · Area_Chair_bPzt · 2022-09-09

**Recommendation:** Accept
**Confidence:** 3

**Metareview:**

Overall, this paper provides a great starting point for future benchmarking experiments. The reviewers engaged in a lively discussion with the authors and provided valuable suggestions for future improvements, which the authors have integrated in their submission.

---

### Decision · Program_Chairs · 2022-09-16

Accept